# Learning with Protection:
# Rejection of Suspicious Samples
# under Adversarial Environment

## Abstract

We propose a novel framework for avoiding the misclassification of data by using a framework of *learning with rejection* and *adversarial examples*. Recent developments in machine learning have opened new opportunities for industrial innovations such as self-driving cars. However, many machine learning models are vulnerable to adversarial attacks and industrial practitioners are concerned about accidents arising from misclassification. To avoid critical misclassifications, we define a sample that is likely to be mislabeled as a suspicious sample. Our main idea is to apply a framework of learning with rejection and adversarial examples to assist in the decision making for such suspicious samples. We propose two frameworks, learning with rejection under adversarial attacks and learning with protection. Learning with rejection under adversarial attacks is a naive extension of the learning with rejection framework for handling adversarial examples. Learning with protection is a practical application of learning with rejection under adversarial attacks. This algorithm transforms the original multi-class classification problem into a binary classification for a specific class, and we reject suspicious samples to protect a specific label. We demonstrate the effectiveness of the proposed method in experiments.

## 1 Introduction

Recent machine learning models have achieved great success in applications such as self-driving cars (Bojarski et al., 2016) and medical diagnosis (Litjens et al., 2017). In a classification problem, we sometimes face the situation where the misclassification of a specific label can result in serious accidents. This problem arises in various situations, and below we provide examples.

**Example 1** (Object Detection for Self-Driving Cars). For self-driving cars, object detection is crucial. For protecting the life of pedestrians, self-driving cars need to be able to detect human or suspend decision making if it detects an object that might be a human.

**Example 2** (Medical Diagnosis). Cancer is one of the leading causes of death globally, and machine learning algorithms for detecting cancer from images have been gaining attention. This is done by detecting abnormalities in an image that might indicate cancer.

As shown above, we want the labeling of data to be conservative for certain situations; for example, in autonomous driving, it is better to misclassify a non-human object as a human than to risk misclassifying a human. In this paper, we introduce two novel frameworks called *learning with rejection under adversarial attacks* and *learning with protection* to protect a specific class from critical misclassification. Our algorithm is based on the techniques of *learning with rejection* and *adversarial examples*. Learning with rejection under adversarial attacks is a naive extension of learning with rejection to hold a decision when we face a suspicious sample that is vulnerable to adversarial attack. Learning with protection is an application of learning with rejection under adversarial attack. The purpose is to protect a specific class from misclassification. Both frameworks are based on the existing work of learning with rejection and adversarial examples.

Learning with rejection is a classification scenario where the learner is given the option to reject an instance instead of predicting its label. The purpose of this framework is to prevent critical

misclassification, where rejecting labeling a datum incurs a lower cost than misclassification. In this field, considerable theoretical and empirical analysis has been conducted (Bartlett & Wegkamp, 2008; Cortes et al., 2016a;b; Grandvalet et al., 2009; Herbei & Wegkamp, 2006; Yuan & Wegkamp, 2010). Learning with rejection is one ultimate solution against data with high uncertainty. However, the existing frameworks have not considered adversarial examples, which refers to a well-designed attack to mislead people to misclassification.

In the industrial application of machine learning, it has been observed that many machine learning models are vulnerable to adversarial attacks. Threat model of adversarial attacks specifies the capabilities of the adversary and classifies adversary attacks to two types of attacks, white-box attack and black-box attack. Adversarial inputs that result in machine learning models returning incorrect outputs are called *adversarial examples*. Several previous studies have artificially generated adversarial examples by adding small perturbations that are imperceptible to humans (Goodfellow et al., 2015; Gu & Rigazio, 2014; Huang et al., 2015; Carlini & Wagner, 2017). Methods for protecting against these adversarial examples are also being proposed. Among them, adversarial training is one of the most effective defenses (Szegedy et al., 2014; Goodfellow et al., 2015; Shaham et al., 2015; Carlini & Wagner, 2016; Papernot et al., 2016; Xu et al., 2017; Madry et al., 2018; Buckman et al., 2018; Kannan et al., 2018; Pang et al., 2018; Wong & Kolter, 2018; Tramèr et al., 2018).

However, most of the defense method fails to avoid misclassification under adversarial attacks. Carlini & Wagner (2017) defeated representative methods for detection of adversarial examples. Athalye et al. (2018) reports that several defense algorithms for white-box setting fail when the attacker uses a carefully designed gradient-based method. Besides, Shafahi et al. (2019); Gilmer et al. (2019) shows that adversarial examples are inevitable in some cases. However, we sometimes face situations in which misclassification can result in fatal accidents. For that purpose, we propose learning with rejection option under adversarial attack. The main idea of our method is to hold a decision when we get suspicious samples. As an application of learning with a rejection option, we propose a method for preventing the misclassification of data for a specific label under adversarial examples. This method considers situations where we must protect our model from adversarial examples by any means. In addition, It should be noted that our method can be combined with other existing methods such as adversarial training. In the following section, we describe the problem setting. In Sections 3 and 4, we propose and describe our algorithm, and in Section 5, we show the experimental results.

## 2 PROBLEM SETTING

We use the standard settings for binary classification problems, i.e., training and test data points are i.i.d. and sampled from some unknown distribution $\mathcal{D}$ over $\mathcal{X} \times \{-1, +1\}$, where $\mathcal{X}$ denotes the input space. The goal is to learn a classifier $h : \mathcal{X} \to \{-1, +1\}$ that assigns a label $\hat{y}(\boldsymbol{x})$ to a datum $\boldsymbol{x}$ as $\hat{y}(\boldsymbol{x}) = \text{sign}(h(\boldsymbol{x}))$. Let $\mathcal{L}$ be a loss function $\mathcal{L} : \mathbb{R} \times \{-1, +1\} \to \mathbb{R}$. The optimal classifier $h^*$ is given by $h^* = \arg\min_{h \in \mathcal{F}} \mathbb{E}[\mathcal{L}(h(X), Y)]$, where $\mathcal{F}$ is a set of measurable functions.

When we train a classifier with training samples, we can naively replace the expectation with the corresponding sample averages. For a hypothesis set $\mathcal{H}$, let us define an estimator of the optimal classifier $h^*$ as $\hat{h} = \arg\min_{h \in \mathcal{H}} \hat{\mathbb{E}}[\mathcal{L}(h(X), Y)]$, where $\hat{\mathbb{E}}$ denotes the averaging operator over training data. Although an estimator $\hat{h}$ converges to $h^*$ in many cases with infinite samples, an estimator $\hat{h}$ might return result that differ greatly from $h^*$ in a case with finite samples. In addition, when $h(\boldsymbol{x})$ is not smooth at around $\boldsymbol{x}$, it might become difficult to estimate the function. These can cause serious problems in real-world applications, such as traffic accidents arising from misclassification by the algorithms used in self-driving cars. To make our algorithm's inference more robust, we allow a learner to reject such suspicious samples. Let ® denote rejection. For any given instance $\boldsymbol{x} \in \mathcal{X}$, the learner has the option of abstaining from assigning a label or rejecting that instance and returning the symbol ®, or assigning the label $\hat{y} \in \{-1, +1\}$. If the learner rejects an instance, then they incur some loss $c(\boldsymbol{x}) \in \mathbb{R}$; if it does not reject but assigns an incorrect label, then it incurs a cost of one; otherwise, it suffers no loss. Thus, the learner outputs a rejection function $r : \mathcal{X} \to \mathbb{R}$, that determines the points $\boldsymbol{x} \in \mathcal{X}$ to be rejected according to $r(\boldsymbol{x}) \leq 0$. Let us denote a loss function with rejection option $\mathcal{L}^{\circledR} : \mathbb{R} \times \{-1, +1\} \to \mathbb{R}$. For example, a loss function can be defined as follows:

$$\mathcal{L}^{\circledR}(h(\boldsymbol{x}), y) = \mathbb{1}_{yh(\boldsymbol{x}) \leq 0} \mathbb{1}_{r(\boldsymbol{x}) \geq 0} + c(\boldsymbol{x}) \mathbb{1}_{r(\boldsymbol{x}) \leq 0}.$$

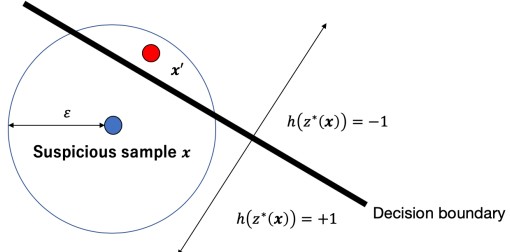

Figure 1: suspicious samples: suspicious samples have a risk of misclassification against the perturbations in $\ell_p$ ball.

There are two frameworks used in the learning with rejection model: the *confidence-based* and *separation-based* approaches Cortes et al. (2016a). The separation-based approach is first formulated by Cortes et al. (2016a). This formulation is a generalization of confidence-based approach (Bartlett & Wegkamp, 2008). In the separation-based approach, we train a classifier and a rejection function simultaneously. In confidence-based approach, given the conditional probability $p(y = +1|\boldsymbol{x})$ and a cost function $c(\boldsymbol{x})$, we only train a classifier and obtain a rejection function automatically after training a classifier. Although the confidence-based approach is a special case of the separation-based approach, the method of confidence-based approach is easy to implement. In this paper, we follow the confidence-based approach and correspond the cost function to vulnerability for adversarial examples.

Our goal is to obtain the optimal classifier $h^*$, which minimizes the classification risk with rejection option, i.e., $h^* = \arg\min_{h \in \mathcal{H}} \mathbb{E}[\mathcal{L}^{\circledR}(h(X), Y)]$.

## 3 LEARNING WITH REJECTION UNDER ADVERSARIAL ATTACKS

In this section, we describe a method to hold a decision against suspicious samples.

### 3.1 STRATEGY FOR REJECTING SUSPICIOUS SAMPLES

To reject suspicious samples, we define which samples are suspicious. We introduce a concept of a *p-norm suspicious sample*. Let $\| \cdot \|_p$ be $\ell_p$-norm. We refer a sample $\boldsymbol{x}$ as suspicious based on the value of $\ell_p$-norm if there exits a sample $\boldsymbol{x}'$ such that $\|\boldsymbol{x} - \boldsymbol{x}'\|_p \leq \varepsilon$ and $\hat{h}(\boldsymbol{x}')$ returns a different class from $\hat{h}(\boldsymbol{x})$. If a sample is suspicious, we refrain from assigning a class label.

Suppose that we have a estimator $\hat{f}(\boldsymbol{x})$ of the class conditional probability $\eta(\boldsymbol{x}) = p(y = +1|\boldsymbol{x})$ and a classifier $\hat{h}(\boldsymbol{x}) = \mathrm{sign}\left(\hat{f}(\boldsymbol{x}) - \frac{1}{2}\right)$. Let $\mathbb{B}_{\boldsymbol{x}}^{\infty}(\epsilon)$ denote the $\ell_p$ ball centered at $\boldsymbol{x} \in \mathcal{X}$ with radius $\epsilon$, i.e., $\mathbb{B}_{\boldsymbol{x}}^p(\epsilon) = \{\boldsymbol{x}' \in \mathcal{X} : \|\boldsymbol{x}' - \boldsymbol{x}\|_p \leq \epsilon\}$. We calculate $\boldsymbol{z}^*(\boldsymbol{x}) \in \mathbb{B}_{\boldsymbol{x}}^p(\epsilon)$ as follows:

$$\boldsymbol{z}^*(\boldsymbol{x}) = \arg\max_{\boldsymbol{x}' \in \mathbb{B}_{\boldsymbol{x}}^p(\epsilon)} \tilde{\mathcal{L}}(\hat{h}(\boldsymbol{x}'), \boldsymbol{x}).$$

By defining $\boldsymbol{z}^*(\boldsymbol{x})$ as in the above equation, we can easily calculate it with the techniques used in adversarial examples. If $\hat{h}(\boldsymbol{z}^*(\boldsymbol{x}))\hat{h}(\boldsymbol{x}) \leq 0$ for a sample $\boldsymbol{x}$, we regard the sample as suspicious one. We illustrate the concept in Figure 1.

### 3.2 RELATIONSHIP WITH LEARNING WITH REJECTION

To interpret the above strategy, we consider the relationship between the strategy and learning with rejection. First, we consider the ideal situation, where we know the true value of the conditional class probability $\eta(\boldsymbol{x}) = p(y = +1|\boldsymbol{x})$. In confidence-based approach, we can determine the rejection function $r(\boldsymbol{x})$ corresponding to the rejection cost as shown by Cortes et al. (2016a). It is known that the classifier $h^*$ defined for any $\boldsymbol{x} \in \mathcal{X}$ by $h^*(\boldsymbol{x}) = \eta(\boldsymbol{x}) - \frac{1}{2}$ is optimal. For any $\boldsymbol{x} \in \mathcal{X}$, the misclassification cost for $h^*$ is $\mathbb{E}\left[\mathbb{1}_{yh(\boldsymbol{x}) \leq 0}|\boldsymbol{x}\right] = \min\{\eta(\boldsymbol{x}), 1 - \eta(\boldsymbol{x})\}$. The optimal rejection

function $r^*(\boldsymbol{x})$ should therefore be defined such that $r^*(\boldsymbol{x}) \leq 0$ if and only if

$$\min\{\eta(\boldsymbol{x}), 1 - \eta(\boldsymbol{x})\} \geq c(\boldsymbol{x}) \Leftrightarrow 1 - \max\{\eta(\boldsymbol{x}), 1 - \eta(\boldsymbol{x})\} \geq c(\boldsymbol{x}) \Leftrightarrow |h(\boldsymbol{x})| \leq \frac{1}{2} - c(\boldsymbol{x}).$$

Therefore, the optimal rejection function $r^*(\boldsymbol{x})$ is given as $r^*(\boldsymbol{x}) = |\eta(\boldsymbol{x}) - \frac{1}{2}| - \left(\frac{1}{2} - c(\boldsymbol{x})\right)$. Thus, in confidence-based approach, we can determine the optimal rejection function $r^*(\boldsymbol{x})$ if we know the conditional probability $\eta(\boldsymbol{x})$ and the rejection cost $c(\boldsymbol{x})$.

To reject suspicious samples, we relate the rejection cost $c(\boldsymbol{x})$ with uncertainty of data $\boldsymbol{x}$. One naive idea is to set $c(\boldsymbol{x})$ as a monotonically decreasing function representing the smoothness of $\eta(\boldsymbol{x})$. When $|\eta(\boldsymbol{x}) - \eta(\boldsymbol{x}')|$ is zero for $\boldsymbol{x}, \boldsymbol{x}' \in \mathbb{R}^d$, $c(\boldsymbol{x})$ takes its largest value; when $\max |f(\boldsymbol{x}) - f(\boldsymbol{x}')|$ is large, $c(\boldsymbol{x})$ is small. This means that, if a sample $\boldsymbol{x}$ is suspicious, then the cost of rejection is low, and we can reject the sample easily; conversely, if a sample $\boldsymbol{x}$ is not suspicious, then the cost of rejection is high. When the cost is high, we hesitate to reject the sample. We assumed this condition could be met when designing our algorithm. As we show later, we can easily find such $c(\boldsymbol{x})$ and reject suspicious samples without expending much energy in determining the value of $c(\boldsymbol{x})$. In our strategy, we reject a sample $\boldsymbol{x}$ such that

$$\left(\eta(\boldsymbol{z}^*(\boldsymbol{x})) - \frac{1}{2}\right)\left(\eta(\boldsymbol{x}) - \frac{1}{2}\right) \leq 0.$$

Therefore, $r^*(\boldsymbol{x}) \leq 0$ if and only if

$$\left(\eta(\boldsymbol{z}^*(\boldsymbol{x})) \leq \frac{1}{2} \text{ and } \eta(\boldsymbol{x}) \geq \frac{1}{2}\right) \text{ or } \left(\eta(\boldsymbol{z}^*(\boldsymbol{x})) \geq \frac{1}{2} \text{ and } \eta(\boldsymbol{x}) \leq \frac{1}{2}\right)$$

$$\Leftrightarrow \left(\eta(\boldsymbol{x}) - \eta(\boldsymbol{x}) + \eta(\boldsymbol{z}^*(\boldsymbol{x})) \leq \frac{1}{2} \text{ and } \eta(\boldsymbol{x}) \geq \frac{1}{2}\right) \text{ or } \left(\eta(\boldsymbol{x}) - \eta(\boldsymbol{x}) + \eta(\boldsymbol{z}^*(\boldsymbol{x})) \geq \frac{1}{2} \text{ and } \eta(\boldsymbol{x}) \leq \frac{1}{2}\right)$$

$$\Leftrightarrow \left(0 \leq \eta(\boldsymbol{x}) - \frac{1}{2} \leq \eta(\boldsymbol{x}) - \eta(\boldsymbol{z}^*(\boldsymbol{x}))\right) \text{ or } \left(\eta(\boldsymbol{x}) - \eta(\boldsymbol{z}^*(\boldsymbol{x})) \leq \eta(\boldsymbol{x}) - \frac{1}{2} \leq 0\right) \qquad (1)$$

Let us assume that $\eta(\boldsymbol{x}) - \eta(\boldsymbol{z}^*(\boldsymbol{x})) \geq 0$ when $\eta(\boldsymbol{x}) - \frac{1}{2} \geq 0$ and $\eta(\boldsymbol{x}) - \eta(\boldsymbol{z}^*(\boldsymbol{x})) \leq 0$ when $\eta(\boldsymbol{x}) - \frac{1}{2} \leq 0$. Then, (1) insists that $r^*(\boldsymbol{x}) \leq 0$ if and only if $|h(\boldsymbol{x})| \leq |\eta(\boldsymbol{x}) - \eta(\boldsymbol{z}^*(\boldsymbol{x}))|$. Therefore, the following relationship holds for a sample $\boldsymbol{x}$ such that $|\eta(\boldsymbol{x}) - \eta(\boldsymbol{z}^*(\boldsymbol{x}))| \leq \frac{1}{2}$:

$$c(\boldsymbol{x}) = \frac{1}{2} - |\eta(\boldsymbol{x}) - \eta(\boldsymbol{z}^*(\boldsymbol{x}))|.$$

Thus, in our strategy, the cost function is a decreasing function of the smoothness of $\eta(\boldsymbol{x})$.

Next, we show a relationship between the cost function and the degree of perturbation $\varepsilon$. Let us assume $\eta(\boldsymbol{x})$ is Lipschitz continuous, with $\ell_p$-norm, and

$$|\eta(\boldsymbol{x}) - \eta(\boldsymbol{x}')| \leq \lambda \|\boldsymbol{x} - \boldsymbol{x}'\|_p = \lambda \varepsilon,$$

where $\boldsymbol{x} \in \mathcal{X}$, $\boldsymbol{x}' \in \mathbb{B}_{\boldsymbol{x}}^p(\epsilon)$, and $\lambda > 0$ is a positive constant. From Lipschitz continuity, $c(\boldsymbol{x}) \geq \frac{1}{2} - \lambda \varepsilon$ holds. This means that the cost of rejection will be small when $\varepsilon$ is a large value. Therefore, a learner can reject a sample with low cost when the learner is afraid of the misclassification and set the possible perturbation of $\boldsymbol{x}$ with $\ell_p$-norm large.

### 3.3 ALGORITHM

Based on the above idea, we develop an algorithm for rejecting suspicious samples. To discuss learning the rejection in the framework of confidence-based approach, we construct an estimator $\hat{f}$ of $\eta(\boldsymbol{x}) = p(y = +1|\boldsymbol{x})$ and a classifier $\hat{h}(\boldsymbol{x}) = \hat{f}(\boldsymbol{x}) - \frac{1}{2}$. Then, we define *pseudo loss function* $\tilde{\mathcal{L}}$ for two samples $\boldsymbol{x} \in \mathcal{X}$ and $\boldsymbol{x}' \in \mathbb{B}_{\boldsymbol{x}}^p(\epsilon)$ as a binary loss function for a pair of $(\boldsymbol{x}', \hat{y}(\boldsymbol{x}))$, where $\hat{y}(\boldsymbol{x}) = \text{sign}(\hat{h}(\boldsymbol{x}))$. For example, if we assume logistic loss, it can be defined as

$$\tilde{\mathcal{L}}(\hat{f}(\boldsymbol{x}'), \hat{y}(\boldsymbol{x})) = -\mathbb{1}_{\hat{y}(\boldsymbol{x})=-1} \log\left(\hat{f}(\boldsymbol{x})\right) - \mathbb{1}_{\hat{y}(\boldsymbol{x})=+1} \log\left(1 - \hat{f}(\boldsymbol{x})\right).$$

Using the pseudo loss, we calculate $\boldsymbol{z}^*(\boldsymbol{x})$ for a sample $\boldsymbol{x}$ as follows:

$$\boldsymbol{z}^*(\boldsymbol{x}) = \underset{\boldsymbol{x}' \in \mathbb{B}_{\boldsymbol{x}}^p(\epsilon)}{\arg\max} \tilde{\mathcal{L}}(\hat{h}(\boldsymbol{x}'), \hat{y}(\boldsymbol{x})).$$

Because this optimization is hard to calculate, we apply techniques of adversarial examples as a heuristics. Then, we reject a sample $\boldsymbol{x}$ if and only if $\hat{h}(\boldsymbol{z}^*(\boldsymbol{x}))\hat{h}(\boldsymbol{x}) \leq 0$.

**Calculation of Adversarial Examples:** Here, we describe the two simple first-order methods for calculating adversarial examples used in this paper. Although various types of adversarial examples have been proposed, hereinafter adversarial examples refer to perturbation-based methods.

One of the simplest methods for generating adversarial examples is the fast gradient sign method (FGSM) Goodfellow et al. (2015), which is a fast single-step attack that maximizes the loss function in the linear approximation. The perturbation under the FGSM is calculated as follows:
$$\boldsymbol{\delta} = \varepsilon \cdot \text{sign}(\nabla_{\boldsymbol{x}} \mathcal{L}(h(\boldsymbol{x}), y)).$$

The projected gradient descent method (PGD) Madry et al. (2018) is an iterative variant of the FGSM. The perturbation under the PGD at time step $s+1$ is calculated as follows:
$$\boldsymbol{\delta}^{(s+1)} = \mathcal{P}_{\varepsilon}(\boldsymbol{\delta}^{(s)} + \alpha \cdot \text{sign}(\nabla_{\boldsymbol{x}} \mathcal{L}(h(\boldsymbol{x} + \boldsymbol{\delta}^{(s)}), t))),$$
where $\alpha$ denotes a single step and $\mathcal{P}_{\varepsilon}(\cdot)$ denotes the projection onto the $\ell_p$-ball with radius $\varepsilon$.

## 4  LEARNING WITH PROTECTION

In this section, we propose a practical and easily implemented algorithm as an application of learning with rejection under adversarial attack. In many cases, we want to avoid misclassification for a specific class. In self-driving cars, we must be able to detect human beings to avoid serious accidents. In medical diagnosis, we should not miss a dangerous disease. We thus propose an algorithm to protect a specific class from misclassification in a multi-class classification problem.

We extend the previous problem setting of binary classification to that of multi-class classification. Let $\mathcal{X}$ and $\mathcal{Y}$ be the feature and label spaces, respectively, and suppose that there is an unknown distribution $\mathcal{D}$ over $\mathcal{X} \times \mathcal{Y}$. We assume that the label space has $K \geq 2$ labels, i.e., $\mathcal{Y} = \{1, 2, 3, ..., K\}$. We define the class for which we want to avoid misclassification as the *defense target class* $t \in \mathcal{Y}$. For example, if we want to avoid misclassification of 1, we designate the defense target class as $t = 1$. We call the algorithm for protecting a defense target class based on the following idea learning with protection.

### 4.1  PROTECTING A DEFENSE TARGET CLASS

Let $\mathcal{F}$ be the set of measurable functions from $\mathcal{X} \times \mathcal{Y}$ to $\mathbb{R}$. For a function $f \in \mathcal{F}$, we define a classifier $h_f : \mathcal{X} \times \mathcal{Y} \to \mathbb{R}$ as follows:
$$y_f(\boldsymbol{x}) = \underset{k \in [K]}{\arg\max} f(\boldsymbol{x}, y = k).$$
Let $\mathcal{L}(f; \boldsymbol{x}, y)$ be the loss function of a function $f \in \mathcal{F}$ for data $(\boldsymbol{x}, y)$. The goal of the learning problem is to find $f \in \mathcal{F}$ such that the population risk $\mathcal{R}(f) := \mathbb{E}_{(\boldsymbol{x}, y) \in \mathcal{D}}[\mathcal{L}(f; \boldsymbol{x}, y)]$ is minimized. This formulation is the standard setting of multi-class classification problem.

From the original formulation of the multi-class classification problem, we construct a binary classification problem to protect the defense target class $t$ from misclassification. Let us define a function $y_f^b : \mathcal{X} \to \{-1, +1\}$ for a function $f$ to transform the prediction of multi-class classification to that of binary classification as
$$y_f^b(\boldsymbol{x}) = \begin{cases} +1 & y_f(\boldsymbol{x}) = t \\ -1 & y_f(\boldsymbol{x}) \neq t \end{cases}.$$
For a pair $(\boldsymbol{x}, y_f^b(\boldsymbol{x}))$, we calculate a new feature $\boldsymbol{z}^*(\boldsymbol{x})$ as follows:
$$\boldsymbol{z}^*(\boldsymbol{x}) = \underset{\boldsymbol{x}' \in \mathbb{B}_{\boldsymbol{x}}^p(\epsilon)}{\arg\max} \tilde{\mathcal{L}}(f; \boldsymbol{x}', y_f^b(\boldsymbol{x})), \tag{2}$$
where $\tilde{\mathcal{L}}_f(f; \boldsymbol{x}', y_f^b(\boldsymbol{x}))$ is a binary loss function for a pair $(\boldsymbol{x}, y_f^b(\boldsymbol{x}))$. This loss function corresponds to the pseudo loss function introduced in the previous section. For example, we can define 0-1 loss as follows:
$$\tilde{\mathcal{L}}(f; \boldsymbol{x}', y_f^b(\boldsymbol{x})) = \mathbb{1}_{y_f^b(\boldsymbol{x}) = -1} \mathbb{1}_{y_f^b(\boldsymbol{x}') = +1} + \mathbb{1}_{y_f^b(\boldsymbol{x}) = +1} \mathbb{1}_{y_f^b(\boldsymbol{x}') = -1}.$$
Then, for this binary classification problem derived from multi-class classification problem, we apply an algorithm for rejection. If $y_f(\boldsymbol{z}^*(\boldsymbol{x})) = t$ for a sample $\boldsymbol{x}$ such that $y_f(\boldsymbol{x}) \neq t$, then we reject the sample because the sample $\boldsymbol{x}$ predicted as a class that is not $t$ might be a class $t$ in neighborhood of $\boldsymbol{x}$ with $\ell_p$-norm. We call algorithms based on the abeve idea *learning with protection*.

---

**Algorithm 1** Learning with Protection

---

**Input:** Trained classier $\hat{h}$, defense target class $t \in [K]$, and test dataset $\{\boldsymbol{x}_i\}_{i=1}^n$.
Construct a pseudo loss function (3) for test dataset $\{\boldsymbol{x}_i\}_{i=1}^n$.
Compute $\boldsymbol{z}^*(\boldsymbol{x}_i)$ for a sample $\boldsymbol{x}_i$ by (2) by method of adversarial example.
Reject a sample $\boldsymbol{x}_i$ if $\hat{h}(\boldsymbol{x}_i) \neq t$ and $\hat{h}(\boldsymbol{z}^*(\boldsymbol{x}_i)) = t$

---

### 4.2 ALGORITHM WITH CROSS-ENTROPY LOSS

Various algorithms can be used to implement the above framework. In this section, we consider a cross-entropy loss function and define an algorithm for protecting the defense target class.

Let $\mathcal{H}$ be a hypothesis set. First, we train a model $f \in \mathcal{H}$ with the cross-entropy loss by any of the suitable methods for classification problems such as adversarial training. After training $g$, we obtain $\hat{f}$, the function that minimizes the cross-entropy loss. The trained model is an estimator of the risk minimizer $f^*$ on population. Second, we use the logistic loss for $\tilde{\mathcal{L}}$, i.e.,

$$
\tilde{\mathcal{L}}(f; \boldsymbol{x}', y_f^b(\boldsymbol{x})) = - \mathbb{1}_{y_f^b(\boldsymbol{x})=-1} \log \left( \frac{\exp\left(f(\boldsymbol{x}', y=t)\right)}{\sum_{k\in[K]} \exp\left(f(\boldsymbol{x}', y=k)\right)} \right)
$$
$$
- \mathbb{1}_{y_f^b(\boldsymbol{x})=+1} \log \left( 1 - \frac{\exp\left(f(\boldsymbol{x}', y=t)\right)}{\sum_{k\in[K]} \exp\left(f(\boldsymbol{x}', y=k)\right)} \right). \tag{3}
$$

The intuition of this loss function is as follows. Let $g^*(\boldsymbol{x}', t) = \frac{\exp\left(f^*(\boldsymbol{x}', y=t)\right)}{\sum_{k\in[K]} \exp(f^*(\boldsymbol{x}', y=k))}$. Under a cross entropy loss function, we can interpret $g^*(\boldsymbol{x}', k)$ as $p(y = k|\boldsymbol{x}')$ with $g^*$. Therefore, we can apply the log loss function function with $g^*(\boldsymbol{x}', t) = p(y = t|\boldsymbol{z})$ and $p(y \neq t|\boldsymbol{z}) = 1 - p(y = t|\boldsymbol{z}) = 1 - g^*(\boldsymbol{x}', t)$. As mentioned above, to find a feature $\boldsymbol{z}^*$ from (2), we can apply methods of adversarial examples as a heuristics. We show the pseudo algorithm in Algorithm 1.

## 5 EXPERIMENTS

To demonstrate the effectiveness of proposed method, we conducted experiments using benchmark data. For the benchmark data, we use the CIFAR-10[1] (Krizhevsky, 2009) and ImageNet-100[2] (Kang et al., 2019) dataset. The CIFAR-10 dataset consists of $32 \times 32$ color images in 10 classes, with $6,000$ images per class. There are $50,000$ training images and $10,000$ test images. The ImageNet-100 dataset consists of the 100-class subset of ImageNet-1K (Deng et al., 2009; Russakovsky et al., 2015) containing every 10th class by WordNet ID order. As we explained above, learning with protection rejects suspicious samples after we train a model $\hat{f}$ to protect defense target class. For CIFAR-10, we set *airplane* class as defense target class. For ImageNet-A, we set the first 10 classes in the order of WordNet ID as defense target class. Our method allow various training method to estimate $f^*$. When we train the model $\hat{f}$, we used both standard training method and adversarial training (Madry et al., 2018). After trainning a model, we adopt 30-step $\ell_\infty$ bounded PGD attack to find the feature $\boldsymbol{z}^*$. For each experiment, we output the *accuracy*, *precision*, *recall*, *rejection rate*, and *true rejection*. Let TP, TN, FP, and FN be the numbers of true positive, true negative, false positive, and false negative of a classification result. Then, accuracy, precision, and recall are defined as accuracy $= \frac{\text{TP+TN}}{\text{TP+TN+FP+FN}}$, precision $= \frac{\text{TP}}{\text{TP+FP}}$, and recall $= \frac{\text{TP}}{\text{TP+FN}}$. Let true accept (TA) be an outcome where the rejection function correctly accepts data such that a sample $\boldsymbol{x}$ does not belong to the defense target class, i.e., $h(\boldsymbol{x}) \neq t$. Let true reject (FR) be an outcome where the rejection function correctly rejects data such that a sample $\boldsymbol{x}$ belongs to the defense target class, i.e., $h(\boldsymbol{x}) = t$. FA and FR can be defined similarly (Ni et al., 2019). Then, rejection rate and precision of rejection are defined as rejection rate $= \frac{\text{TR+FR}}{\text{TR+TA+FR+FA}}$ and precision of rejection $= \frac{\text{TR}}{\text{TR+FR}}$.

---

[1]See https://www.cs.toronto.edu/~kriz/cifar.html.
[2]See http://image-net.org/index.

Table 1: Results of adversarial trained model against $\ell_\infty$ bounded PGD attack. Results of standard trained model are shown in Appendix B.1. We show the accuracy (Acc), precision (Pre), recall (Rec), rejection rate (Rej), and precision of rejection (PR).

| | | CIFAR-10 | | | | | | | ImageNet-100 | | | | |
| | | Metric | | | | | | | Metric | | | | |
| Attack | Protection | Acc | Pre | Rec | Rej | PR | Attack | Protection | Acc | Pre | Rec | Rej | PR |
|---|---|---|---|---|---|---|---|---|---|---|---|---|---|
| clean | without | 98.15 | 94.05 | 87.00 | 0.00 | 0.00 | clean | without | 97.72 | 88.14 | 89.20 | 0.00 | 0.00 |
| | 4/255 | 99.18 | 94.05 | 97.42 | 04.66 | 22.96 | | 4/255 | 98.49 | 88.14 | 98.67 | 12.30 | 7.74 |
| | 16/255 | 97.40 | 94.05 | 100.0 | 78.86 | 1.65 | | 16/255 | 91.50 | 88.14 | 100.0 | 85.20 | 1.26 |
| 4/255 | without | 94.80 | 80.61 | 63.20 | 0.00 | 0.00 | 4/255 | without | 94.16 | 71.76 | 68.60 | 0.00 | 0.00 |
| | 4/255 | 98.13 | 80.61 | 97.53 | 10.07 | 34.96 | | 4/255 | 96.60 | 71.76 | 98.00 | 16.37 | 18.18 |
| | 16/255 | 91.03 | 80.61 | 100.0 | 83.06 | 4.43 | | 16/255 | 79.48 | 71.76 | 100.0 | 86.15 | 3.62 |
| 16/255 | without | 84.94 | 12.01 | 8.00 | 0.00 | 0.00 | 16/255 | without | 88.34 | 40.10 | 33.60 | 0.00 | 0.00 |
| | 4/255 | 85.67 | 12.01 | 8.93 | 2.15 | 48.37 | | 4/255 | 88.80 | 40.10 | 35.59 | 0.91 | 60.87 |
| | 16/255 | 46.92 | 12.01 | 100.0 | 88.96 | 10.34 | | 16/255 | 61.66 | 40.10 | 96.00 | 85.85 | 7.51 |

**Neural network model:** For CIFAR-10, we use the ResNet-56 (He et al., 2015) architecture. For ImageNet-100, we use the ResNet-50 (He et al., 2015) architecture with $224 \times 224$ resolution as implemented in torchvision. We describe training hyperparameters in Appendix A.

**Adversarial training:** We use $\ell_\infty$ bounded projected gradient descent (PGD) method to generate training samples. We select a attack target class for each image uniformly at random from the set of incorrect classes. For distortion size $\varepsilon$, we set as $\varepsilon = 8/255$ and apply random scaling $\text{Uniform}(0, \varepsilon)$ to improve performance against smaller distortions. We use $10$ optimization steps and step size $\varepsilon/\sqrt{\text{steps}}$ as described in (Kang et al., 2019). We update the model by stochastic gradient descent (SGD) method using only the adversarial images (no clean images).

## 5.1 PROTECTION AGAINST ARTIFICIAL ADVERSARIAL EXAMPLES

We evaluate our learning with protection method on the CIFAR-10 and ImageNet-100 validation sets against artificial adversarial examples with the defense target class.

**protection against expected attack:** We distort the inputs by 50-step $\ell_\infty$ bounded PGD attack and apply learning with protection method. The single step size of PGD is calculated by $\varepsilon/\sqrt{\text{steps}}$. The results are shown in Table 1 and Appendix B.1.

**protection against unexpected attack:** Although in our learning with protection method, a specific adversarial attack method is used to find the feature $z^*(x_i)$ for a sample $x_i$ (we adopt $\ell_\infty$ bounded PGD in this experiment), the input might be distorted by different attack methods. In order to evaluate the performance against such unexpected adversaries, we conducted experiments against 50-step $l_2$ bounded PGD attack. The results are shown in Table 2 and Appendix B.1.

We can see from the table that our method achieves $100\%$ in the recall, i.e., we can reject all samples such that $y = t$ and $h(x) \neq t$. As shown in Appendix B.1, models with adversarial training outperforms models without adversarial training. It is because our method only reduce the false negative, and it is difficult to improve the performance of a model when the number of the true positive is small. When we use adversarial training, we can increase the recall with the small number of rejection. For example, in Table 2, when attack is $\ell_2$ bounded PGD ($\varepsilon = 80$) for CIFAR-10, recall increases about $20\%$ by only reject $9.14\%$ of all samples. On the other hand, On the other hand, when we reject too many samples, the precision of rejection dramatically drops. Therefore, we need to control the perturbation $\varepsilon$ not to reject samples unnecessarily.

## 5.2 PROTECTION AGAINST NATURAL ADVERSARIAL EXAMPLES

Natural adversarial examples which are introduced in (Hendrycks et al., 2019) are defined as unmodified and naturally occurring examples that cause classifier accuracy to degrade drastically. In order to evaluate the proposed method for such real hard samples, we conduct experiments on ImageNet-A[3]

---

[3] See https://github.com/hendrycks/natural-adv-examples.

Table 2: Results of adversarial trained model against $\ell_2$ bounded PGD attack. Results of standard trained model are shown in Appendix B.1. We show the accuracy (Acc), precision (Pre), recall (Rec), rejection rate (Rej), and precision of rejection (PR).

| | | CIFAR-10 | | | | | | | ImageNet-100 | | | | |
|---|---|---|---|---|---|---|---|---|---|---|---|---|---|
| Attack | Protection | | Metric | | | | Attack | Protection | | Metric | | | |
| | | Acc | Pre | Rec | Rej | PR | | | Acc | Pre | Rec | Rej | PR |
| clean | without | 98.15 | 94.05 | 87.00 | 0.00 | 0.00 | clean | without | 97.72 | 88.14 | 89.20 | 0.00 | 0.00 |
| | 4/255 | 99.18 | 94.05 | 97.42 | 04.66 | 22.96 | | 4/255 | 98.49 | 88.14 | 98.67 | 12.30 | 7.74 |
| | 16/255 | 97.40 | 94.05 | 100.0 | 78.86 | 1.65 | | 16/255 | 91.50 | 88.14 | 100.0 | 85.20 | 1.26 |
| 80 | without | 95.56 | 84.49 | 68.10 | 0.00 | 0.00 | 600 | without | 93.18 | 66.46 | 64.20 | 0.00 | 0.00 |
| | 4/255 | 98.45 | 84.49 | 97.70 | 9.14 | 33.15 | | 4/255 | 95.60 | 66.46 | 93.86 | 16.67 | 18.81 |
| | 16/255 | 93.02 | 84.49 | 100.0 | 82.10 | 3.89 | | 16/255 | 75.42 | 66.46 | 100.0 | 86.13 | 4.12 |
| 320 | without | 85.10 | 16.16 | 11.70 | 0.00 | 0.00 | 2400 | without | 87.78 | 37.24 | 32.40 | 0.00 | 0.00 |
| | 4/255 | 86.27 | 16.16 | 14.10 | 3.84 | 44.27 | | 4/255 | 88.24 | 37.24 | 34.25 | 0.65 | 81.82 |
| | 16/255 | 55.24 | 16.16 | 100.0 | 86.44 | 10.22 | | 16/255 | 56.47 | 37.24 | 96.43 | 86.49 | 7.62 |

Table 3: Results of ImageNet-A. Full experimental results are shown in Appendix B.2. We show the accuracy (Acc), precision (Pre), recall (Rec), rejection rate (Rej), and precision of rejection (PR).

| | Adversarial Trained | | | | | Standard Trained | | | | |
|---|---|---|---|---|---|---|---|---|---|---|
| Protection | | Metric | | | | | Metric | | | |
| | Acc | Pre | Rec | Rej | PR | Acc | Pre | Rec | Rej | PR |
| without | 76.84 | 39.98 | 21.78 | 00.00 | 00.00 | 77.69 | 42.88 | 20.56 | 00.00 | 00.00 |
| $\varepsilon = 2/255$ | 80.23 | 39.98 | 39.15 | 29.72 | 31.18 | 42.88 | 42.88 | 100.0 | 89.99 | 18.43 |
| $\varepsilon = 4/255$ | 78.46 | 39.98 | 72.40 | 60.27 | 24.23 | 42.88 | 42.88 | 100.0 | 89.99 | 18.43 |
| $\varepsilon = 8/255$ | 52.41 | 39.98 | 99.42 | 85.60 | 19.05 | 42.88 | 42.88 | 100.0 | 89.99 | 18.43 |

dataset. ImageNet-A contains 7,500 natural adversarial example images in 200 classes that are a subset of ImageNet-1K's 1,000 classes. We use the ResNet-50 architecture with $224 \times 224$ resolution. We first trained the model on the training set of ImageNet-1K dataset and evaluate the performance on ImageNet-A with our learning with protection method at test time. Due to the difference in the number of classes, only 200 of its 1,000 logits are used in test time. We set the first 20 classes in the order of WordNet ID as the defense target class. The results are shown in Table 3.

As shown in Table 3 and Appendix B.2, the proposed method works well for natural adversarial samples. For example, when the distortion size is $2/255$, the recall increases about $20\%$ by rejecting only $30\%$ of all samples. On the other hand, the rejection rate of the models without adversarial training is high even when the size of the perturbation $\varepsilon$ is small.

## 6 CONCLUSION

In this paper, we proposed learning with rejection under adversarial attacks for avoiding the misclassification. The proposed method suspend decision making when a sample is $\ell_p$-norm suspicious. By extending the strategy, we proposed an easily implementable algorithm called learning with protection. This algorithm prevents a defense target class from misclassification. We showed the performance of learning with protection using benchmark datasets. From the results, we confirmed that our method successfully caught samples that would have misclassified if we did not reject it.

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

# A TRAINING HYPERPARAMETERS

For CIFAR-10, we trained on a single NVIDIA V100 GPU for 200 epochs with batch size 32, initial learning rate 0.1, momentum 0.9, and weight decay $10^{-4}$. We decayed the learning rate at epochs 100 and 150. For ImageNet-100, we trained on machines with 8 NVIDIA V100 GPUs using standard data augmentation used in (He et al., 2015). We ran synchronized SGD (Goyal et al., 2017) for 90 epochs with batch size $32 \times 8$ and a learning rate schedule with 5 "warm-up" epochs and a decay at epochs 30, 60, and 80 by a factor of 10.

# B FULL EXPERIMENTAL RESULTS

## B.1 PROTECTION AGAINST ARTIFICIAL ADVERSARIAL EXAMPLES

Table 4: Results of adversarial trained ResNet-56 against $\ell_\infty$ bounded PGD attack for CIFAR-10.

| Attack | Protection | Metric | | | | |
|---|---|---|---|---|---|---|
| | | Acc [%] | Pre [%] | Rec [%] | Rej [%] | PR[%] |
| clean | without | 98.15 | 94.05 | 87.00 | 0.00 | 0.00 |
| | $\varepsilon = 2/255$ | 98.71 | 94.05 | 92.36 | 01.58 | 36.71 |
| | $\varepsilon = 4/255$ | 99.18 | 94.05 | 97.42 | 04.66 | 22.96 |
| | $\varepsilon = 8/255$ | 99.26 | 94.05 | 99.66 | 22.08 | 5.75 |
| | $\varepsilon = 16/255$ | 97.40 | 94.05 | 100.0 | 78.86 | 1.65 |
| | $\varepsilon = 32/255$ | 94.08 | 94.05 | 100.0 | 90.71 | 1.43 |
| $\varepsilon = 2/255$ | without | 97.16 | 91.15 | 79.30 | 0.00 | 0.00 |
| | $\varepsilon = 2/255$ | 98.73 | 91.15 | 94.52 | 3.45 | 46.67 |
| | $\varepsilon = 4/255$ | 98.98 | 91.15 | 97.90 | 7.42 | 25.61 |
| | $\varepsilon = 8/255$ | 98.93 | 91.15 | 99.75 | 25.98 | 7.89 |
| | $\varepsilon = 16/255$ | 96.00 | 91.15 | 100.0 | 80.73 | 2.56 |
| | $\varepsilon = 32/255$ | 91.17 | 91.15 | 100.0 | 91.28 | 2.27 |
| $\varepsilon = 4/255$ | without | 94.80 | 80.61 | 63.20 | 0.00 | 0.00 |
| | $\varepsilon = 2/255$ | 96.71 | 80.61 | 79.50 | 4.14 | 49.52 |
| | $\varepsilon = 4/255$ | 98.13 | 80.61 | 97.53 | 10.07 | 34.96 |
| | $\varepsilon = 8/255$ | 97.83 | 80.61 | 100.0 | 29.95 | 12.29 |
| | $\varepsilon = 16/255$ | 91.03 | 80.61 | 100.0 | 83.06 | 4.43 |
| | $\varepsilon = 32/255$ | 80.64 | 80.61 | 100.0 | 92.15 | 3.99 |
| $\varepsilon = 8/255$ | without | 89.22 | 44.90 | 34.30 | 0.00 | 0.00 |
| | $\varepsilon = 2/255$ | 90.63 | 44.90 | 41.13 | 2.70 | 61.48 |
| | $\varepsilon = 4/255$ | 92.48 | 44.90 | 56.05 | 8.30 | 46.75 |
| | $\varepsilon = 8/255$ | 93.59 | 44.90 | 100.0 | 34.34 | 19.13 |
| | $\varepsilon = 16/255$ | 70.80 | 44.90 | 100.0 | 85.58 | 7.68 |
| | $\varepsilon = 32/255$ | 45.04 | 44.90 | 100.0 | 92.34 | 7.12 |
| $\varepsilon = 16/255$ | without | 84.94 | 12.01 | 8.00 | 0.00 | 0.00 |
| | $\varepsilon = 2/255$ | 85.13 | 12.01 | 8.21 | 0.37 | 67.57 |
| | $\varepsilon = 4/255$ | 85.67 | 12.01 | 8.93 | 2.15 | 48.37 |
| | $\varepsilon = 8/255$ | 87.58 | 12.01 | 20.00 | 27.05 | 22.18 |
| | $\varepsilon = 16/255$ | 46.92 | 12.01 | 100.0 | 88.96 | 10.34 |
| | $\varepsilon = 32/255$ | 12.01 | 12.01 | 100.0 | 93.34 | 9.86 |
| $\varepsilon = 32/255$ | without | 86.11 | 10.06 | 4.90 | 0.00 | 0.00 |
| | $\varepsilon = 2/255$ | 86.11 | 10.06 | 4.90 | 0.00 | 0.00 |
| | $\varepsilon = 4/255$ | 86.10 | 10.06 | 4.90 | 0.16 | 6.25 |
| | $\varepsilon = 8/255$ | 85.56 | 10.06 | 4.99 | 5.08 | 3.54 |
| | $\varepsilon = 16/255$ | 72.16 | 10.06 | 24.87 | 78.95 | 10.17 |
| | $\varepsilon = 32/255$ | 10.04 | 10.06 | 98.00 | 95.12 | 9.99 |

Table 5: Results of standard trained ResNet-56 against $\ell_\infty$ bounded PGD attack for CIFAR-10.

| Attack | Protection | Metric | | | | |
|---|---|---|---|---|---|---|
| | | Acc [%] | Pre [%] | Rec [%] | Rej [%] | PR[%] |
| clean | without | 98.78 | 94.34 | 93.40 | 0.00 | 0.00 |
| | $\varepsilon = 2/255$ | 97.34 | 94.34 | 100.0 | 78.96 | 0.84 |
| | $\varepsilon = 4/255$ | 94.42 | 94.34 | 100.0 | 89.96 | 0.73 |
| | $\varepsilon = 8/255$ | 94.36 | 94.34 | 100.0 | 90.07 | 0.73 |
| | $\varepsilon = 16/255$ | 94.64 | 94.34 | 100.0 | 89.56 | 0.74 |
| | $\varepsilon = 32/255$ | 95.17 | 94.34 | 99.89 | 88.20 | 0.74 |
| $\varepsilon = 2/255$ | without | 82.41 | 5.61 | 4.80 | 0.00 | 0.00 |
| | $\varepsilon = 2/255$ | 62.60 | 5.61 | 64.86 | 77.73 | 11.91 |
| | $\varepsilon = 4/255$ | 32.92 | 5.61 | 96.00 | 87.94 | 10.80 |
| | $\varepsilon = 8/255$ | 10.36 | 5.61 | 88.89 | 90.93 | 10.40 |
| | $\varepsilon = 16/255$ | 13.39 | 5.61 | 54.55 | 90.22 | 10.11 |
| | $\varepsilon = 32/255$ | 21.23 | 5.61 | 47.06 | 89.07 | 10.08 |
| $\varepsilon = 4/255$ | without | 83.68 | 4.86 | 3.40 | 0.00 | 0.00 |
| | $\varepsilon = 2/255$ | 79.72 | 4.86 | 5.30 | 37.19 | 9.63 |
| | $\varepsilon = 4/255$ | 78.75 | 4.86 | 8.15 | 50.64 | 11.51 |
| | $\varepsilon = 8/255$ | 64.85 | 4.86 | 40.96 | 79.66 | 11.51 |
| | $\varepsilon = 16/255$ | 7.11 | 4.86 | 100.0 | 92.83 | 10.41 |
| | $\varepsilon = 32/255$ | 4.86 | 4.86 | 100.0 | 93.00 | 10.39 |
| $\varepsilon = 8/255$ | without | 86.39 | 5.65 | 2.30 | 0.00 | 0.00 |
| | $\varepsilon = 2/255$ | 85.66 | 5.65 | 2.35 | 6.70 | 3.43 |
| | $\varepsilon = 4/255$ | 85.44 | 5.65 | 2.42 | 9.81 | 4.89 |
| | $\varepsilon = 8/255$ | 85.15 | 5.65 | 2.96 | 23.36 | 9.55 |
| | $\varepsilon = 16/255$ | 71.01 | 5.65 | 12.99 | 81.44 | 10.11 |
| | $\varepsilon = 32/255$ | 53.43 | 5.65 | 13.61 | 88.62 | 9.38 |
| $\varepsilon = 16/255$ | without | 88.10 | 7.59 | 1.70 | 0.00 | 0.00 |
| | $\varepsilon = 2/255$ | 87.97 | 7.59 | 1.71 | 1.32 | 2.27 |
| | $\varepsilon = 4/255$ | 87.89 | 7.59 | 1.71 | 2.25 | 2.67 |
| | $\varepsilon = 8/255$ | 87.84 | 7.59 | 1.72 | 3.09 | 3.88 |
| | $\varepsilon = 16/255$ | 86.69 | 7.59 | 1.92 | 19.23 | 5.98 |
| | $\varepsilon = 32/255$ | 70.54 | 7.59 | 8.29 | 86.59 | 9.18 |
| $\varepsilon = 32/255$ | without | 89.10 | 5.88 | 0.60 | 0.00 | 0.00 |
| | $\varepsilon = 2/255$ | 89.10 | 5.88 | 0.60 | 0.03 | 0.00 |
| | $\varepsilon = 4/255$ | 89.04 | 5.88 | 0.60 | 0.95 | 4.21 |
| | $\varepsilon = 8/255$ | 89.02 | 5.88 | 0.60 | 1.11 | 3.60 |
| | $\varepsilon = 16/255$ | 88.94 | 5.88 | 0.60 | 1.97 | 3.05 |
| | $\varepsilon = 32/255$ | 85.72 | 5.88 | 0.63 | 27.37 | 1.94 |

Table 6: Results of adversarial trained ResNet-56 against $\ell_2$ bounded PGD attack for CIFAR-10.

| Attack | Protection | Metric | | | | |
|---|---|---|---|---|---|---|
| | | Acc [%] | Pre [%] | Rec [%] | Rej [%] | PR[%] |
| clean | without | 98.15 | 94.05 | 87.00 | 0.00 | 0.00 |
| | $\varepsilon = 2/255$ | 98.71 | 94.05 | 92.36 | 01.58 | 36.71 |
| | $\varepsilon = 4/255$ | 99.18 | 94.05 | 97.42 | 04.66 | 22.96 |
| | $\varepsilon = 8/255$ | 99.26 | 94.05 | 99.66 | 22.08 | 5.75 |
| | $\varepsilon = 16/255$ | 97.40 | 94.05 | 100.0 | 78.86 | 1.65 |
| | $\varepsilon = 32/255$ | 94.08 | 94.05 | 100.0 | 90.71 | 1.43 |
| $\varepsilon = 40$ | without | 97.38 | 91.55 | 81.30 | 0.00 | 0.00 |
| | $\varepsilon = 2/255$ | 98.71 | 91.55 | 94.21 | 2.97 | 46.13 |
| | $\varepsilon = 4/255$ | 99.01 | 91.55 | 97.95 | 6.79 | 25.04 |
| | $\varepsilon = 8/2550$ | 98.96 | 91.55 | 99.63 | 24.94 | 7.38 |
| | $\varepsilon = 16/255$ | 96.20 | 91.55 | 100.0 | 80.26 | 2.33 |
| | $\varepsilon = 32/255$ | 91.57 | 91.55 | 100.0 | 91.10 | 2.05 |
| $\varepsilon = 80$ | without | 95.56 | 84.49 | 68.10 | 0.00 | 0.00 |
| | $\varepsilon = 2/255$ | 97.59 | 84.49 | 86.53 | 4.19 | 50.84 |
| | $\varepsilon = 4/255$ | 98.45 | 84.49 | 97.70 | 9.14 | 33.15 |
| | $\varepsilon = 8/2550$ | 98.26 | 84.49 | 100.0 | 28.12 | 11.34 |
| | $\varepsilon = 16/255$ | 93.02 | 84.49 | 100.0 | 82.10 | 3.89 |
| | $\varepsilon = 32/255$ | 84.55 | 84.49 | 100.0 | 91.91 | 3.47 |
| $\varepsilon = 160$ | without | 90.73 | 54.75 | 42.10 | 0.00 | 0.00 |
| | $\varepsilon = 2/255$ | 92.28 | 54.75 | 51.53 | 3.67 | 49.86 |
| | $\varepsilon = 4/255$ | 94.52 | 54.75 | 74.25 | 9.83 | 44.05 |
| | $\varepsilon = 8/2550$ | 94.80 | 54.75 | 98.83 | 32.12 | 17.87 |
| | $\varepsilon = 16/255$ | 78.33 | 54.75 | 100.0 | 83.94 | 6.90 |
| | $\varepsilon = 32/255$ | 54.81 | 54.75 | 100.0 | 92.30 | 6.27 |
| $\varepsilon = 320$ | without | 85.10 | 16.16 | 11.70 | 0.00 | 0.00 |
| | $\varepsilon = 2/255$ | 85.44 | 16.16 | 12.25 | 0.75 | 60.00 |
| | $\varepsilon = 4/255$ | 86.27 | 16.16 | 14.10 | 3.84 | 44.27 |
| | $\varepsilon = 8/2550$ | 88.80 | 16.16 | 42.86 | 31.85 | 22.83 |
| | $\varepsilon = 16/255$ | 55.24 | 16.16 | 100.0 | 86.44 | 10.22 |
| | $\varepsilon = 32/255$ | 16.28 | 16.16 | 100.0 | 92.75 | 9.52 |
| $\varepsilon = 640$ | without | 84.80 | 9.63 | 6.20 | 0.00 | 0.00 |
| | $\varepsilon = 2/255$ | 84.80 | 9.63 | 6.20 | 0.01 | 0.00 |
| | $\varepsilon = 4/255$ | 84.76 | 9.63 | 6.20 | 0.28 | 0.00 |
| | $\varepsilon = 8/2550$ | 83.63 | 9.63 | 6.60 | 10.90 | 5.60 |
| | $\varepsilon = 16/255$ | 60.27 | 9.63 | 66.67 | 84.57 | 10.72 |
| | $\varepsilon = 32/255$ | 09.63 | 9.63 | 100.0 | 93.56 | 10.03 |

Table 7: Results of standard trained ResNet-56 against $\ell_2$ bounded PGD attack for CIFAR-10.

| Attack | Protection | Metric | | | | |
|---|---|---|---|---|---|---|
| | | Acc [%] | Pre [%] | Rec [%] | Rej [%] | PR[%] |
| clean | without | 98.78 | 94.34 | 93.40 | 0.00 | 0.00 |
| | $\varepsilon = 2/255$ | 97.34 | 94.34 | 100.0 | 78.96 | 0.84 |
| | $\varepsilon = 4/255$ | 94.42 | 94.34 | 100.0 | 89.96 | 0.73 |
| | $\varepsilon = 8/255$ | 94.36 | 94.34 | 100.0 | 90.07 | 0.73 |
| | $\varepsilon = 16/255$ | 94.64 | 94.34 | 100.0 | 89.56 | 0.74 |
| | $\varepsilon = 32/255$ | 95.17 | 94.34 | 99.89 | 88.20 | 0.74 |
| $\varepsilon = 40$ | without | 83.98 | 12.09 | 09.60 | 0.00 | 0.00 |
| | $\varepsilon = 2/255$ | 54.26 | 12.09 | 100.0 | 84.74 | 10.67 |
| | $\varepsilon = 4/255$ | 13.61 | 12.09 | 100.0 | 91.92 | 9.83 |
| | $\varepsilon = 8/2550$ | 12.20 | 12.09 | 100.0 | 92.05 | 9.82 |
| | $\varepsilon = 16/255$ | 12.09 | 12.09 | 100.0 | 92.06 | 9.82 |
| | $\varepsilon = 32/255$ | 12.09 | 12.09 | 100.0 | 92.06 | 9.82 |
| $\varepsilon = 80$ | without | 82.74 | 5.62 | 04.60 | 0.00 | 0.00 |
| | $\varepsilon = 2/255$ | 68.30 | 5.62 | 42.99 | 73.72 | 12.11 |
| | $\varepsilon = 4/255$ | 44.40 | 5.62 | 86.79 | 85.99 | 11.01 |
| | $\varepsilon = 8/2550$ | 12.39 | 5.62 | 77.97 | 91.04 | 10.34 |
| | $\varepsilon = 16/255$ | 15.12 | 5.62 | 43.81 | 90.21 | 9.92 |
| | $\varepsilon = 32/255$ | 24.13 | 5.62 | 38.33 | 88.85 | 9.90 |
| $\varepsilon = 160$ | without | 84.72 | 5.10 | 3.00 | 0.00 | 0.00 |
| | $\varepsilon = 2/255$ | 81.63 | 5.10 | 3.70 | 27.12 | 6.97 |
| | $\varepsilon = 4/255$ | 81.41 | 5.10 | 4.91 | 38.74 | 10.04 |
| | $\varepsilon = 8/255$ | 74.74 | 5.10 | 20.27 | 73.24 | 11.63 |
| | $\varepsilon = 16/255$ | 11.85 | 5.10 | 66.67 | 93.50 | 10.21 |
| | $\varepsilon = 32/255$ | 8.45 | 5.10 | 65.22 | 93.73 | 10.18 |
| $\varepsilon = 320$ | without | 86.90 | 6.21 | 2.20 | 0.00 | 0.00 |
| | $\varepsilon = 2/255$ | 86.55 | 6.21 | 2.22 | 3.36 | 2.98 |
| | $\varepsilon = 4/255$ | 86.31 | 6.21 | 2.23 | 5.16 | 2.33 |
| | $\varepsilon = 8/255$ | 85.87 | 6.21 | 2.37 | 12.40 | 5.81 |
| | $\varepsilon = 16/255$ | 79.10 | 6.21 | 6.77 | 69.62 | 9.70 |
| | $\varepsilon = 32/255$ | 57.63 | 6.21 | 11.11 | 88.01 | 9.11 |
| $\varepsilon = 640$ | without | 88.24 | 5.10 | 1.00 | 0.00 | 0.00 |
| | $\varepsilon = 2/255$ | 88.24 | 5.10 | 1.01 | 0.45 | 11.11 |
| | $\varepsilon = 4/255$ | 88.16 | 5.10 | 1.01 | 1.29 | 5.43 |
| | $\varepsilon = 8/255$ | 88.14 | 5.10 | 1.01 | 1.63 | 5.52 |
| | $\varepsilon = 16/255$ | 87.62 | 5.10 | 1.03 | 7.67 | 4.30 |
| | $\varepsilon = 32/255$ | 74.15 | 5.10 | 4.13 | 83.83 | 9.04 |

Table 8: Results of adversarial trained ResNet-50 against $\ell_\infty$ bounded PGD attack for ImageNet-100.

| Attack | Protection | Metric | | | | |
|---|---|---|---|---|---|---|
| | | Acc [%] | Pre [%] | Rec [%] | Rej [%] | PR[%] |
| clean | without | 97.72 | 88.14 | 89.20 | 0.00 | 0.00 |
| | $\varepsilon = 2/255$ | 98.31 | 88.14 | 95.50 | 3.97 | 16.50 |
| | $\varepsilon = 4/255$ | 98.49 | 88.14 | 98.67 | 12.30 | 7.74 |
| | $\varepsilon = 8/255$ | 97.78 | 88.14 | 100.0 | 45.54 | 2.35 |
| | $\varepsilon = 16/255$ | 91.50 | 88.14 | 100.0 | 85.20 | 1.26 |
| | $\varepsilon = 32/255$ | 88.64 | 88.14 | 100.0 | 88.73 | 1.21 |
| $\varepsilon = 2/255$ | without | 96.36 | 82.72 | 80.40 | 0.00 | 0.00 |
| | $\varepsilon = 2/255$ | 97.89 | 82.72 | 96.40 | 6.07 | 27.12 |
| | $\varepsilon = 4/255$ | 97.91 | 82.72 | 99.01 | 15.79 | 11.81 |
| | $\varepsilon = 8/2550$ | 96.65 | 82.72 | 100.0 | 49.40 | 3.94 |
| | $\varepsilon = 16/255$ | 87.79 | 82.72 | 100.0 | 85.56 | 2.27 |
| | $\varepsilon = 32/255$ | 83.63 | 82.72 | 100.0 | 89.03 | 2.18 |
| $\varepsilon = 4/255$ | without | 94.16 | 71.76 | 68.60 | 0.00 | 0.00 |
| | $\varepsilon = 2/255$ | 95.63 | 71.76 | 82.06 | 3.89 | 41.84 |
| | $\varepsilon = 4/255$ | 96.60 | 71.76 | 98.00 | 16.37 | 18.18 |
| | $\varepsilon = 8/2550$ | 94.26 | 71.76 | 99.71 | 52.18 | 5.93 |
| | $\varepsilon = 16/255$ | 79.48 | 71.76 | 100.0 | 86.15 | 3.62 |
| | $\varepsilon = 32/255$ | 73.53 | 71.76 | 100.0 | 89.09 | 3.50 |
| $\varepsilon = 8/255$ | without | 90.68 | 53.76 | 48.60 | 0.00 | 0.00 |
| | $\varepsilon = 2/255$ | 91.73 | 53.76 | 55.10 | 1.57 | 74.68 |
| | $\varepsilon = 4/255$ | 93.24 | 53.76 | 70.23 | 7.62 | 40.10 |
| | $\varepsilon = 8/255$ | 90.91 | 53.76 | 99.59 | 53.39 | 9.51 |
| | $\varepsilon = 16/255$ | 64.64 | 53.76 | 100.0 | 87.48 | 5.83 |
| | $\varepsilon = 32/255$ | 57.26 | 53.76 | 100.0 | 89.50 | 5.70 |
| $\varepsilon = 16/255$ | without | 88.34 | 40.10 | 33.60 | 0.00 | 0.00 |
| | $\varepsilon = 2/255$ | 88.43 | 40.10 | 33.94 | 0.10 | 100.0 |
| | $\varepsilon = 4/255$ | 88.80 | 40.10 | 35.59 | 0.91 | 60.87 |
| | $\varepsilon = 8/255$ | 90.00 | 40.10 | 69.42 | 34.72 | 14.74 |
| | $\varepsilon = 16/255$ | 61.66 | 40.10 | 96.00 | 85.85 | 7.51 |
| | $\varepsilon = 32/255$ | 45.84 | 40.10 | 98.25 | 89.90 | 7.26 |
| $\varepsilon = 32/255$ | without | 86.56 | 33.14 | 33.80 | 0.00 | 0.00 |
| | $\varepsilon = 2/255$ | 86.56 | 33.14 | 33.80 | 0.02 | 0.00 |
| | $\varepsilon = 4/255$ | 86.55 | 33.14 | 33.80 | 0.10 | 0.00 |
| | $\varepsilon = 8/255$ | 86.57 | 33.14 | 35.58 | 3.63 | 13.66 |
| | $\varepsilon = 16/255$ | 85.30 | 33.14 | 40.62 | 19.86 | 8.39 |
| | $\varepsilon = 32/255$ | 83.76 | 33.14 | 45.80 | 33.12 | 7.85 |

Table 9: Results of standard trained ResNet-50 against $\ell_\infty$ bounded PGD attack for ImageNet-100.

| Attack | Protection | Metric | | | | |
|---|---|---|---|---|---|---|
| | | Acc [%] | Pre [%] | Rec [%] | Rej [%] | PR[%] |
| clean | without | 98.80 | 94.00 | 94.00 | 0.00 | 0.00 |
| | $\varepsilon = 2/255$ | 94.59 | 94.00 | 100.0 | 88.19 | 0.67 |
| | $\varepsilon = 4/255$ | 94.12 | 94.00 | 100.0 | 89.09 | 0.67 |
| | $\varepsilon = 8/255$ | 94.09 | 94.00 | 100.0 | 89.13 | 0.67 |
| | $\varepsilon = 16/255$ | 94.04 | 94.00 | 100.0 | 89.23 | 0.67 |
| | $\varepsilon = 32/255$ | 94.00 | 94.00 | 100.0 | 89.29 | 0.67 |
| $\varepsilon = 2/255$ | without | 88.88 | 43.49 | 37.40 | 0.00 | 0.00 |
| | $\varepsilon = 2/255$ | 86.89 | 43.49 | 68.25 | 49.25 | 9.11 |
| | $\varepsilon = 4/255$ | 84.06 | 43.49 | 77.27 | 62.12 | 8.24 |
| | $\varepsilon = 8/2550$ | 66.31 | 43.49 | 95.41 | 84.37 | 7.15 |
| | $\varepsilon = 16/255$ | 44.27 | 43.49 | 100.0 | 90.56 | 6.86 |
| | $\varepsilon = 32/255$ | 43.49 | 43.49 | 100.0 | 90.67 | 6.85 |
| $\varepsilon = 4/255$ | without | 88.76 | 42.51 | 35.20 | 0.00 | 0.00 |
| | $\varepsilon = 2/255$ | 88.54 | 42.51 | 36.97 | 6.07 | 7.84 |
| | $\varepsilon = 4/255$ | 88.65 | 42.51 | 38.60 | 8.63 | 10.11 |
| | $\varepsilon = 8/2550$ | 88.46 | 42.51 | 46.32 | 23.19 | 10.27 |
| | $\varepsilon = 16/255$ | 71.37 | 42.51 | 88.89 | 81.19 | 7.38 |
| | $\varepsilon = 32/255$ | 42.51 | 42.51 | 100.0 | 90.99 | 7.06 |
| $\varepsilon = 8/255$ | without | 88.28 | 39.61 | 32.80 | 0.00 | 0.00 |
| | $\varepsilon = 2/255$ | 88.25 | 39.61 | 32.87 | 0.44 | 4.55 |
| | $\varepsilon = 4/255$ | 88.26 | 39.61 | 32.93 | 0.54 | 7.41 |
| | $\varepsilon = 8/255$ | 88.19 | 39.61 | 33.00 | 1.29 | 4.62 |
| | $\varepsilon = 16/255$ | 87.47 | 39.61 | 38.23 | 17.68 | 7.97 |
| | $\varepsilon = 32/255$ | 49.20 | 39.61 | 97.62 | 89.29 | 7.38 |
| $\varepsilon = 16/255$ | without | 86.92 | 32.89 | 29.60 | 0.00 | 0.00 |
| | $\varepsilon = 2/255$ | 86.91 | 32.89 | 29.60 | 0.08 | 0.00 |
| | $\varepsilon = 4/255$ | 86.91 | 32.89 | 29.60 | 0.08 | 0.00 |
| | $\varepsilon = 8/255$ | 86.91 | 32.89 | 29.60 | 0.08 | 0.00 |
| | $\varepsilon = 16/255$ | 86.81 | 32.89 | 29.66 | 1.01 | 1.96 |
| | $\varepsilon = 32/255$ | 82.84 | 32.89 | 44.85 | 43.23 | 7.80 |
| $\varepsilon = 32/255$ | without | 83.34 | 23.53 | 29.60 | 0.00 | 0.00 |
| | $\varepsilon = 2/255$ | 83.33 | 23.53 | 29.60 | 0.06 | 0.00 |
| | $\varepsilon = 4/255$ | 83.33 | 23.53 | 29.60 | 0.08 | 0.00 |
| | $\varepsilon = 8/255$ | 83.33 | 23.53 | 29.60 | 0.08 | 0.00 |
| | $\varepsilon = 16/255$ | 83.32 | 23.53 | 29.60 | 0.14 | 0.00 |
| | $\varepsilon = 32/255$ | 83.19 | 23.53 | 29.96 | 1.61 | 7.41 |

Table 10: Results of adversarial trained ResNet-50 against $\ell_2$ bounded PGD attack for ImageNet-100.

| Attack | Protection | Metric | | | | |
|---|---|---|---|---|---|---|
| | | Acc [%] | Pre [%] | Rec [%] | Rej [%] | PR[%] |
| clean | without | 97.72 | 88.14 | 89.20 | 0.00 | 0.00 |
| | $\varepsilon = 2/255$ | 98.31 | 88.14 | 95.50 | 3.97 | 16.50 |
| | $\varepsilon = 4/255$ | 98.49 | 88.14 | 98.67 | 12.30 | 7.74 |
| | $\varepsilon = 8/255$ | 97.78 | 88.14 | 100.0 | 45.54 | 2.35 |
| | $\varepsilon = 16/255$ | 91.50 | 88.14 | 100.0 | 85.20 | 1.26 |
| | $\varepsilon = 32/255$ | 88.64 | 88.14 | 100.0 | 88.73 | 1.21 |
| $\varepsilon = 300$ | without | 95.84 | 79.67 | 78.40 | 0.00 | 0.00 |
| | $\varepsilon = 2/255$ | 97.51 | 79.67 | 95.84 | 6.03 | 29.93 |
| | $\varepsilon = 4/255$ | 97.48 | 79.67 | 98.74 | 16.41 | 12.45 |
| | $\varepsilon = 8/2550$ | 95.98 | 79.67 | 100.0 | 49.90 | 4.29 |
| | $\varepsilon = 16/255$ | 85.55 | 79.67 | 100.0 | 85.48 | 2.51 |
| | $\varepsilon = 32/255$ | 80.73 | 79.67 | 100.0 | 88.91 | 2.41 |
| $\varepsilon = 600$ | without | 93.18 | 66.46 | 64.20 | 0.00 | 0.00 |
| | $\varepsilon = 2/255$ | 94.55 | 66.46 | 76.25 | 3.89 | 40.31 |
| | $\varepsilon = 4/255$ | 95.60 | 66.46 | 93.86 | 16.67 | 18.81 |
| | $\varepsilon = 8/2550$ | 93.10 | 66.46 | 100.0 | 52.62 | 6.75 |
| | $\varepsilon = 16/255$ | 75.42 | 66.46 | 100.0 | 86.13 | 4.12 |
| | $\varepsilon = 32/255$ | 68.54 | 66.46 | 100.0 | 88.99 | 3.99 |
| $\varepsilon = 1200$ | without | 89.74 | 48.57 | 44.20 | 0.00 | 0.00 |
| | $\varepsilon = 2/255$ | 90.36 | 48.57 | 47.73 | 1.21 | 60.66 |
| | $\varepsilon = 4/255$ | 91.57 | 48.57 | 58.62 | 7.36 | 33.15 |
| | $\varepsilon = 8/255$ | 89.90 | 48.57 | 96.09 | 51.47 | 10.41 |
| | $\varepsilon = 16/255$ | 61.06 | 48.57 | 100.0 | 87.28 | 6.34 |
| | $\varepsilon = 32/255$ | 52.54 | 48.57 | 100.0 | 89.42 | 6.19 |
| $\varepsilon = 2400$ | without | 87.78 | 37.24 | 32.40 | 0.00 | 0.00 |
| | $\varepsilon = 2/255$ | 87.90 | 37.24 | 32.86 | 0.14 | 100.0 |
| | $\varepsilon = 4/255$ | 88.24 | 37.24 | 34.25 | 0.65 | 81.82 |
| | $\varepsilon = 8/255$ | 88.48 | 37.24 | 52.77 | 27.22 | 14.07 |
| | $\varepsilon = 16/255$ | 56.47 | 37.24 | 96.43 | 86.49 | 7.62 |
| | $\varepsilon = 32/255$ | 41.40 | 37.24 | 98.18 | 89.86 | 7.40 |
| $\varepsilon = 4800$ | without | 87.10 | 35.23 | 34.60 | 0.00 | 0.00 |
| | $\varepsilon = 2/255$ | 87.10 | 35.23 | 34.60 | 0.00 | 0.00 |
| | $\varepsilon = 4/255$ | 87.10 | 35.23 | 34.60 | 0.00 | 0.00 |
| | $\varepsilon = 8/255$ | 86.91 | 35.23 | 35.16 | 2.62 | 6.06 |
| | $\varepsilon = 16/255$ | 84.77 | 35.23 | 48.32 | 33.69 | 8.36 |
| | $\varepsilon = 32/255$ | 80.51 | 35.23 | 58.45 | 54.31 | 7.45 |

Table 11: Results of standard trained ResNet-50 against $\ell_2$ bounded PGD attack for ImageNet-100.

| Attack | Protection | Metric | | | | |
|---|---|---|---|---|---|---|
| | | Acc [%] | Pre [%] | Rec [%] | Rej [%] | PR[%] |
| clean | without | 98.80 | 94.00 | 94.00 | 0.00 | 0.00 |
| | $\varepsilon = 2/255$ | 94.59 | 94.00 | 100.0 | 88.19 | 0.67 |
| | $\varepsilon = 4/255$ | 94.12 | 94.00 | 100.0 | 89.09 | 0.67 |
| | $\varepsilon = 8/255$ | 94.09 | 94.00 | 100.0 | 89.13 | 0.67 |
| | $\varepsilon = 16/255$ | 94.04 | 94.00 | 100.0 | 89.23 | 0.67 |
| | $\varepsilon = 32/255$ | 94.00 | 94.00 | 100.0 | 89.29 | 0.67 |
| $\varepsilon = 300$ | without | 88.86 | 43.23 | 36.40 | 0.00 | 0.00 |
| | $\varepsilon = 2/255$ | 72.95 | 43.23 | 92.39 | 80.58 | 7.46 |
| | $\varepsilon = 4/255$ | 60.43 | 43.23 | 98.91 | 87.12 | 7.20 |
| | $\varepsilon = 8/2550$ | 46.05 | 43.23 | 100.0 | 90.42 | 6.98 |
| | $\varepsilon = 16/255$ | 43.50 | 43.23 | 100.0 | 90.81 | 6.95 |
| | $\varepsilon = 32/255$ | 43.23 | 43.23 | 100.0 | 90.85 | 6.94 |
| $\varepsilon = 600$ | without | 88.80 | 42.92 | 36.40 | 0.00 | 0.00 |
| | $\varepsilon = 2/255$ | 88.59 | 42.92 | 50.28 | 25.81 | 10.61 |
| | $\varepsilon = 4/255$ | 88.07 | 42.92 | 55.32 | 34.52 | 9.83 |
| | $\varepsilon = 8/2550$ | 81.49 | 42.92 | 77.45 | 67.58 | 7.78 |
| | $\varepsilon = 16/255$ | 45.50 | 42.92 | 100.0 | 90.40 | 6.98 |
| | $\varepsilon = 32/255$ | 42.92 | 42.92 | 100.0 | 90.79 | 6.95 |
| $\varepsilon = 1200$ | without | 88.36 | 40.28 | 34.00 | 0.00 | 0.00 |
| | $\varepsilon = 2/255$ | 88.22 | 40.28 | 34.27 | 1.88 | 4.21 |
| | $\varepsilon = 4/255$ | 88.15 | 40.28 | 34.34 | 2.62 | 3.79 |
| | $\varepsilon = 8/255$ | 88.00 | 40.28 | 35.49 | 6.43 | 6.48 |
| | $\varepsilon = 16/255$ | 82.23 | 40.28 | 61.82 | 59.35 | 7.52 |
| | $\varepsilon = 32/255$ | 40.57 | 40.28 | 100.0 | 90.79 | 7.21 |
| $\varepsilon = 2400$ | without | 87.80 | 37.15 | 31.80 | 0.00 | 0.00 |
| | $\varepsilon = 2/255$ | 87.78 | 37.15 | 31.80 | 0.14 | 0.00 |
| | $\varepsilon = 4/255$ | 87.78 | 37.15 | 31.80 | 0.16 | 0.00 |
| | $\varepsilon = 8/255$ | 87.77 | 37.15 | 31.80 | 0.22 | 0.00 |
| | $\varepsilon = 16/255$ | 87.62 | 37.15 | 32.45 | 3.08 | 6.45 |
| | $\varepsilon = 32/255$ | 68.99 | 37.15 | 80.71 | 79.56 | 7.56 |
| $\varepsilon = 4800$ | without | 85.86 | 29.42 | 29.60 | 0.00 | 0.00 |
| | $\varepsilon = 2/255$ | 85.85 | 29.42 | 29.60 | 0.10 | 0.00 |
| | $\varepsilon = 4/255$ | 85.85 | 29.42 | 29.60 | 0.10 | 0.00 |
| | $\varepsilon = 8/255$ | 85.85 | 29.42 | 29.60 | 0.10 | 0.00 |
| | $\varepsilon = 16/255$ | 85.84 | 29.42 | 29.60 | 0.14 | 0.00 |
| | $\varepsilon = 32/255$ | 85.20 | 29.42 | 30.96 | 7.36 | 5.93 |

## B.2 Protection against Natural Adversarial Examples

Table 12: Results of adversarial trained ResNet-50 for ImageNet-A.

| Protection | Metric | | | | |
|---|---|---|---|---|---|
| | Acc [%] | Pre [%] | Rec [%] | Rej [%] | PR[%] |
| without | 76.84 | 39.98 | 21.78 | 00.00 | 00.00 |
| $\varepsilon = 2/255$ | 80.23 | 39.98 | 39.15 | 29.72 | 31.18 |
| $\varepsilon = 4/255$ | 78.46 | 39.98 | 72.40 | 60.27 | 24.23 |
| $\varepsilon = 8/255$ | 52.41 | 39.98 | 99.42 | 85.60 | 19.05 |
| $\varepsilon = 16/255$ | 40.60 | 39.98 | 100.0 | 88.51 | 18.45 |
| $\varepsilon = 32/255$ | 39.98 | 39.98 | 100.0 | 88.63 | 18.43 |

Table 13: Results of standard trained ResNet-50 for ImageNet-A.

| Protection | Metric | | | | |
|---|---|---|---|---|---|
| | Acc [%] | Pre [%] | Rec [%] | Rej [%] | PR[%] |
| without | 77.69 | 42.88 | 20.56 | 00.00 | 00.00 |
| $\varepsilon = 2/255$ | 42.88 | 42.88 | 100.0 | 89.99 | 18.43 |
| $\varepsilon = 4/255$ | 42.88 | 42.88 | 100.0 | 89.99 | 18.43 |
| $\varepsilon = 8/255$ | 42.88 | 42.88 | 100.0 | 89.99 | 18.43 |
| $\varepsilon = 16/255$ | 42.88 | 42.88 | 100.0 | 89.99 | 18.43 |
| $\varepsilon = 32/255$ | 42.88 | 42.88 | 100.0 | 89.99 | 18.43 |

