# OpenReview forum: "Learning with Protection: Rejection of Suspicious Samples under Adversarial Environment"
_ICLR.cc/2020/Conference — Reject_

### Official Review · AnonReviewer2 · 2019-10-22
**Official Blind Review #2**

**Rating:** 3

**Review:**

This paper wants to study the problem of “learning with rejection under adversarial attacks”. It first naively extends the learning with rejection framework for handling adversarial examples. It then considers the classical cost-sensitive learning by transfer the multi-class problem into binary classification problem through one-vs-all and using the technique they proposed to reject predictions on non-important labels, and name such technique as “learning with protection”. Finally, they do some experimental studies.

The paper does not show any connection between “learning with rejection” and “adversarial learning”. The method it proposes is also a naïve extension of existing methods. Both the problem setting and the technique does not have novelty. The paper fails to realize that the motivated application is actually called “cost-sensitive learning” and has been studied long time before. The paper also has problems in writing. Finally, there is no comparison with any baseline. Only empirical results of the proposed methods are shown. Due to all these reasons, there is still a long way to go before the paper can be published. I will rate it a clear rejection.

More specially,

The definition of “suspicious example” in Sec.3.1 has no relationship with adversary examples. Does the paper focus on adversary examples? If the definition has no relationship, it is classical learning with rejection.
In the last equation of Page 3, there is no definition of \tilde L. Actually, according to Figure 1, x’s is more close to the decision boundary, it is an example more hard to classify, which could also be “suspicious”.
In the definition of “suspicious example” at the beginning of Sec.3.1, is both x and x’ defined as suspicious examples in this way?
In the last equation of page 2, there is a rejection function, so minimizing this loss is a “separation-based approach”. However, at the end of Sec.2 the paper states they “follow a confidence-based approach”. Any comment on the inconsistency?

The motivated problem is not new. It is called cost-sensitive learning in machine learning and can date back to 2001:
Charles Elkan. The Foundations of Cost-Sensitive Learning. IJCAI 2001: 973-978.
Where they study the same problem when misclassifying one class of data may cost a lot than misclassifying another class of data. The current paper has not discussed any related work of cost-sensitive learning although they want to study a problem in its field.

The paper should be also improved in writing in the following aspects.
There is a lot of inaccurate statements in the paper. For example,  “In Sections 3 and 4, we propose and describe our algorithm”, what is the difference between propose and describe? “an estimator \hat h might return result that differ greatly from h^* in a case with finite samples”. Actually there are rigorous theoretical results describing how the number of finite samples will impact the estimator \hat h on unseen data. For example,
Peter L. Bartlett, Shahar Mendelson. Rademacher and Gaussian Complexities: Risk Bounds and Structural Results. JMLR, 2002.
So inaccurate/unclear statements that will mislead readers should be avoided.

In writing, the paper also lacks the necessary references in many places. For example, “Learning with rejection is a classification scenario where the learner is given the option to reject an instance instead of predicting its label.”, “…classifies adversary attacks to two types of attacks, white-box attack and black-box attack.”, “Methods for protecting against these adversarial examples are also being proposed.”. Necessary references are needed for these places.

The organization is also problematic. For example, in the second half of Sec.2 introducing two kinds of learning with rejection models, it should be included in a “related work” part.

------------------------------------------
Thank you for the rebuttal. I raised my score a little bit. But I still think this paper has not been ready to be published yet.


**Experience Assessment:**

I have read many papers in this area.

**Review Assessment: Checking Correctness Of Derivations And Theory:**

N/A

**Review Assessment: Checking Correctness Of Experiments:**

I carefully checked the experiments.

**Review Assessment: Thoroughness In Paper Reading:**

I read the paper thoroughly.

---

> ### Author Response · Authors · 2019-11-15
> **Response to AnonReviewer2**
>
> Dear Reviewer,
>
> Thank you for your comments. Our replies to the comments are listed below.
>
> Q1. The definition of “suspicious example” in Sec.3.1 has no relationship with adversary examples. Does the paper focus on adversary examples? If the definition has no relationship, it is classical learning with rejection.
> A1. This definition is the standard definition of adversarial examples and adversarial attacks. For example, [1] introduced the definition as the traditional definition of adversarial examples. We can derive the definition from the formulation of adversarial training, min_f max_{x'\in B(x)} \ell(f(x), y) [2]. The definition of suspicious samples is based on the traditional definition of adversarial examples, and we extended the traditional definition to introduce the proposed algorithm.
> [1] Theoretical Framework for Robustness of (Deep) Classifiers, ICLR workshop 2017
> [2] Rademacher Complexity for Adversarially Robust Generalization, ICML 2019
>
> Q2. In the last equation of Page 3, there is no definition of \tilde L. Actually, according to Figure 1, x’s is more close to the decision boundary, it is an example more hard to classify, which could also be “suspicious”.
> A2. In page 3, to introduce the idea of the proposed algorithm, we did not explain the definition of \tilde L, and we explained it in Eq.(8). However, as you pointed out, it is confusion and we will revise it in the next division.
>
> Q3. In the definition of “suspicious example” at the beginning of Sec.3.1, is both x and x’ defined as suspicious examples in this way?
> A3. Yes,  both x and x’  can be suspicious examples. However, as we explained in section 4, we can select a specific class that should be protected. If we are only afraid of the misclassification of a  class of x, we define x only in learning with protection.
>
> Q4. In the last equation of page 2, there is a rejection function, so minimizing this loss is a “separation-based approach”. However, at the end of Sec.2 the paper states they “follow a confidence-based approach”. Any comment on the inconsistency?
> A4. It is consistent. The last equation of page2 is not only for “separation-based approach”, but also "confidence-based approach”.
>
> Q5. The motivated problem is not new. It is called cost-sensitive learning in machine learning and can date back to 2001.
> A5. We think that the motivation for cost-sensitive learning is different from that of our research. In cost-sensitive learning, they do not care about the existence of adversarial examples.
>
> Q6. “an estimator \hat h might return result that differ greatly from h^* in a case with finite samples”. Actually there are rigorous theoretical results describing how the number of finite samples will impact the estimator \hat h on unseen data. So inaccurate/unclear statements that will mislead readers should be avoided.
> A6. To motivate our research, we described that the adversarial examples are caused by estimation error. However, as you pointed out, the sentence will confuse readers. We change the expression in the next revision.
>
> Q7. In the last equation of Page 3, there is no definition of \tilde L. In writing, the paper also lacks the necessary references in many places. The organization is also problematic. For example, in the second half of Sec.2 introducing two kinds of learning with rejection models, it should be included in a “related work” part.
> A7. Thank you for pointing out them. We will revise our paper by reflecting your opinions.

---

### Official Review · AnonReviewer1 · 2019-10-23
**Official Blind Review #1**

**Rating:** 3

**Review:**

The paper proposes a framework for learning with rejection using ideas from adversarial examples. The essential idea is, while predicting on a point x, we can reject classifying the point if it has an adversarial example very close to it. So, the algorithm can be simply summarized as,
1. Learn a classifier function f
2. On the test set, predict on a point, only if it doesn't have an adversarial example close by.

I am inclined to reject the paper for the following reasons:
1. The proposed approach is a variation of a fairly well-known heuristic. Having a close adversarial example is same as saying that the current point is very close to the decision boundary. Being close to the decision boundary is a heuristic that has been applied in multiple scenarios in machine learning.
2. The proposed approach is not novel. For example, [1] uses adversarial example style detection to augment their training data and improve their end-to-end model.
3. There have been approaches which attempt to learn rejection function [2], so it would have been good to at least do a comparison of the proposed approach with such methods.

[1] Adversarial Examples For Improving End-to-End Attention-based Small-Footprint Keyword Spotting, ICASSP 2019
[2] SelectiveNet: A Deep Neural Network with an Integrated Reject Option, ICML 2019

---

Thanks for the rebuttal. I have raised my scores, but I still believe that this paper falls short of acceptance.

**Experience Assessment:**

I have read many papers in this area.

**Review Assessment: Checking Correctness Of Derivations And Theory:**

I carefully checked the derivations and theory.

**Review Assessment: Checking Correctness Of Experiments:**

I carefully checked the experiments.

**Review Assessment: Thoroughness In Paper Reading:**

I read the paper thoroughly.

---

> ### Author Response · Authors · 2019-11-15
> **Response to AnonReviewer1**
>
> Dear Reviewer,
>
> Thank you for your insightful comments. Our replies to the comments are listed below.
>
> Q1. The proposed approach is not novel. For example, [1] uses adversarial example style detection to augment their training data and improve their end-to-end model.
> A1. The idea of our paper is different from that of Wang et al. [1]'s paper. They augmented their samples using adversarial examples. On the other hand, we reject some samples that will mislead the learner's decision making. The idea of Wang et al. [1]'s paper is similar to adversarial training. However, it is well known that adversarial training can be a cat‐and‐mouse game, i.e., a model trained by adversarial training is still weak against a different attack that is not assumed in the training process. Because training a robust model is a difficult task as many existing works pointed out, our idea is to propose a new way of decision making to deal with adversarial examples by introducing a rejection option.
>
> Q2. There have been approaches which attempt to learn rejection function [2], so it would have been good to at least do a comparison of the proposed approach with such methods.
> A2. We agree with your opinion. In fact, in the next research, we should consider learning a rejection function and classifier together. However, because the paper [2] does not consider the existence of adversarial examples, we consider that the method proposed in [2] is weak against adversarial examples. Therefore, we did not compare our method with their method that considers standard learning with rejection.

---

### Official Review · AnonReviewer3 · 2019-10-23
**Official Blind Review #3**

**Rating:** 3

**Review:**

There is still no universal method to deal with adversarial examples, and introducing a reject option to flag potential attacks seems a sensitive choice for many applications. The considered problem is well-motivated and introduced, and I’m unaware of prior work studying classification with reject option in the context of adversarial examples. However, I think there are different dimensions along which the paper could be improved:

- My understanding of classification with a reject option is that the rejection cost c(x) is a design choice that can depend on the specific application. While c(x) is introduced as part of the framework it is then derived in a very specific way, removing the design aspect or at least not explaining very clearly how one would design it. Also, Algorithm 1 doesn’t have corresponding parameters.

- The rejection function r(x) relies on z^* which essentially amounts to computing an adversarial perturbation for a given testing point, see (2). The authors state that they use a 30-step of the PGD algorithm to find z^*. The attacks on which the method is tested only uses 10 PGD steps. What if the attacker is stronger in the sense that it runs significantly more PGD iterations than used to compute the rejection function? How sensitive is the rejection function to different initializations?

- Similarly, what happens if the attacker and the rejection function rely on different norms to compute the attack and the rejection score, respectively? I think it is important to investigate this aspect.

- In Tables 1 and 2, I don’t understand why the precision is the same for all rows corresponding to the same attack (strength), while the other metrics vary.

Overall, I think the paper explores an interesting direction, but would greatly benefit from a revision along the lines outlined above.

Minor comments:
- P3 3.1 2nd paragraph: “Let B^p...” rather than “Let B^\infty...”
- P4 bottom: “)” is missing before “..., where”
- P6 bottom: (FR) should be (TR). The acronyms FA and FR don’t seem to be introduced.



###
Reply to rebuttal:

I thank the reviewers for their detailed reply. It seems that the authors agree that some of the issues I raised should be addressed and improved. My assessment that the paper should be revised (and accordingly the rating) therefore remains unchanged.


**Experience Assessment:**

I have read many papers in this area.

**Review Assessment: Checking Correctness Of Derivations And Theory:**

I assessed the sensibility of the derivations and theory.

**Review Assessment: Checking Correctness Of Experiments:**

I assessed the sensibility of the experiments.

**Review Assessment: Thoroughness In Paper Reading:**

I read the paper at least twice and used my best judgement in assessing the paper.

---

> ### Author Response · Authors · 2019-11-15
> **Response to AnonReviewer3**
>
> Dear Reviewer,
>
> Thank you for your inquiry about the proposed method and the experimental results. Our replies to the questions are listed below.
>
> Q1. My understanding of classification with a reject option is that the rejection cost c(x) is a design choice that can depend on the specific application.
> A1. In this paper, we proposed a new framework of applying learning with rejection to adversarial examples. As far as we know, there is no method based on a similar idea. As you pointed out, we also think that we should design our algorithm to enable us to control c(x). However, there are other difficulties in the theoretical justification for it. Therefore, based on this paper, we tackle the problem of designing c(x) in the next research.
>
> Q2. The rejection function r(x) relies on z^* which essentially amounts to computing an adversarial perturbation for a given testing point, see (2). The authors state that they use a 30-step of the PGD algorithm to find z^*. The attacks on which the method is tested only uses 10 PGD steps. What if the attacker is stronger in the sense that it runs significantly more PGD iterations than used to compute the rejection function? How sensitive is the rejection function to different initializations?
> A2. We also think that we need to increase the number of iterations of PGD when an attacker uses a stronger PGD attack. Following your opinion, in the next revision, we will show the sensitivity by experiment.
>
> Q3. Similarly, what happens if the attacker and the rejection function rely on different norms to compute the attack and the rejection score, respectively? I think it is important to investigate this aspect.
> A3. We have shown the results in Table 2. In the experiments, we used l_inf norm to reject suspicious samples and l_2 norm to attack the model. We confirmed that the proposed method is effective against an unexpected attack.
>
> Q4.  In Tables 1 and 2, I don’t understand why the precision is the same for all rows corresponding to the same attack (strength), while the other metrics vary.
> A4. The proposed algorithm rejects suspicious samples. Therefore, it does not change the numbers of samples that are predicted as positive. It means that the numbers of true positive and false positive do not change. Because precision = TP/(TP+FP), precision also does not change. Even though precision does not change, we showed the result to evaluate the algorithm.

---

### Decision · Program_Chairs · 2019-12-19

**Decision:**

Reject

**Comment:**

The paper addresses the setting of learning with rejection while incorporating the ideas from learning with adversarial examples to tackle adversarial attacks. While the reviewers acknowledged the importance to study learning with rejection in this setting, they raised several concerns: (1) lack of technical contribution -- see R1’s and R2’s related references, see R3’s suggestion on designing c(x); (2) insufficient empirical evidence -- see R3’s comment about the sensitivity experiment on the strength of the attack, see R1’s suggestion to compare with a baseline that learns the rejection function such as SelectiveNet;  (3) clarity of presentation -- see R2’s suggestions how to improve clarity.
Among these, (3) did not have a substantial impact on the decision, but would be helpful to address in a subsequent revision. However, (1) and (2) make it very difficult to assess the benefits of the proposed approach, and were viewed by AC as critical issues.
AC can confirm that all three reviewers have read the author responses and have revised the final ratings. AC suggests, in its current state the manuscript is not ready for a publication. We hope the reviews are useful for improving and revising the paper.